# Neutrophils-to-Lymphocyte Ratio Is Associated with Progression and Overall Survival in Amyotrophic Lateral Sclerosis

**DOI:** 10.3390/biomedicines10020354

**Published:** 2022-02-01

**Authors:** Maurizio A. Leone, Jessica Mandrioli, Sergio Russo, Aliona Cucovici, Giulia Gianferrari, Vitalie Lisnic, Dafin Fior Muresanu, Francesco Giuliani, Massimiliano Copetti, Andrea Fontana

**Affiliations:** 1Neurology Unit, Fondazione IRCCS Casa Sollievo della Sofferenza, 71013 San Giovanni Rotondo, Italy; alionacucovici85@gmail.com; 2Department of Biomedical, Metabolic and Neural Sciences, University of Modena and Reggio Emilia, 41121 Modena, Italy; jmandrio@unimore.it (J.M.); gianferrari.giulia@gmail.com (G.G.); 3Neurology Unit, Department of Neurosciences, Azienda Ospedaliero Universitaria di Modena, 41125 Modena, Italy; 4Innovation & Research Unit, Fondazione IRCCS Casa Sollievo della Sofferenza, 71013 San Giovanni Rotondo, Italy; srusso@operapadrepio.it (S.R.); f.giuliani@operapadrepio.it (F.G.); 5Department of Medical and Surgical Sciences, University of Foggia, 71122 Foggia, Italy; 6Department of Neurology, State University of Medicine and Pharmacy “Nicolae Testemitanu”, 2004 Chisinau, Moldova; vitalie.lisnic@usmf.md; 7Department of Clinical Neurosciences, University of Medicine & Pharmacy, 400000 Cluj-Napoca, Romania; dafinm@ssnn.ro; 8“RoNeuro” Institute for Neurological Research and Diagnostic, 400364 Cluj-Napoca, Romania; 9Unit of Biostatistics, Fondazione IRCCS Casa Sollievo della Sofferenza, 71013 San Giovanni Rotondo, Italy; m.copetti@operapadrepio.it (M.C.); a.fontana@operapadrepio.it (A.F.)

**Keywords:** amyotrophic lateral sclerosis, neutrophil-to-lymphocyte ratio, disease progression rate, prognosis, survival, inflammation

## Abstract

Background: Amyotrophic lateral sclerosis (ALS) is a devastating and untreatable motor neuron disease, with a 3–5-year survival from diagnosis. Possible prognostic serum biomarkers include albumin, C-reactive protein, ferritin, creatinine, uric acid, hemoglobin, potassium, sodium, calcium, glucose, and the neutrophil-to-lymphocyte ratio (NLR), a marker of subclinical inflammation. Objective: To ascertain the influence of NLR on ALS progression rate and survival. Methods: Cross-sectional multicenter study including 146 consecutive incident and prevalent patients (88 males), aged >18 years, diagnosed according to the El Escorial criteria. The exclusion criteria were: (1) patients with tracheostomy or receiving mechanical ventilation; (2) patients with percutaneous endoscopic gastrostomy; and (3) patients who did not sign the informed consent. The rate of disease progression (ΔFS score) represents the monthly decline of the ALSFRS-R score, and was computed as (48 − total ALSFRS-R at recruitment)/symptom duration in months. Patients were followed up to tracheotomy, death, or the end of the follow-up, whichever occurred first. To validate our findings, we used data retrieved from the Pooled Resource Open-Access ALS Clinical Trials (PRO-ACT) Database. Results: The median disease duration was 15 (range = 2–30) months. The mean ALSFRS-R score at recruitment was 35.8 ± 8.0 (range: 10–48), and the median ΔFS was 0.66 (range: 0–5.33). Age at onset, at diagnosis, and at recruitment were significantly lower in the lowest NLR tertile. NLR values positively correlated with ΔFS values (r = 0.28): the regression slope of NLR (log-values) was 0.60 (*p* < 0.001) before and 0.49 (*p* = 0.006) after adjustment for age at recruitment. The ΔFS score progressively increased from the lowest to the highest NLR tertile: 0.35 (IQR: 0.18–0.93), 0.62 (IQR: 0.25–1.09), and 0.86 (IQR: 0.53–1.92). Patients were followed for a median of 2 years. The mortality rate passed from 15.9 events per 100 person-years in patients belonging to the lowest NLR tertile to 52.8 in those in the highest tertile. The optimal cut-off value which best classified patients with the lowest and the highest mortality rate was set at the NLR value of 2.315. Indeed, the mortality rate of patients with an NLR value above such cut-off was twice the mortality rate of patients with a value below the cut-off (age adjusted hazard ratio (HR): 2.16, 95% confidence interval (CI): 1.32–3.53). In the PRO-ACT validation sample, patients with an NLR value above the cut-off consistently had a higher mortality rate than those with a value below the cut-off (age adjusted HR: 1.17, 95%CI: 1.01–1.35). Conclusions: NLR could be a candidate easy, fast, and low-cost marker of disease progression and survival in ALS. It may be associated with low-grade inflammation either as a direct mirror of the pathological process of disease progression, or as a consequence of neuronal death (reverse causation). However, prospective studies are needed to understand whether NLR changes during the course of the disease, before using it to monitor disease progression in ALS.

## 1. Introduction

Amyotrophic lateral sclerosis (ALS) is an intractable neurodegenerative disease characterized by progressive loss of upper motor neurons in the motor cortex and lower motor neurons in the brainstem and spinal cord; survival is usually 3–5 years from diagnosis [1]. One of the goals of patient management is to detect prognostic factors as early as possible, in order to give indications to patients and their families and to plan interventions. Prediction of disease progression and survival has so far mainly been based upon clinical features, including age, El-Escorial diagnostic category at diagnosis, forced vital capacity (FVC), site of onset, phenotype, BMI, and diagnostic delay [2,3]. Given the urgent need to find sensitive, reliable, and easy-to-obtain biomarkers for monitoring disease progression and predict prognosis, attention has been recently turned to body fluid biomarkers, such as serum albumin, C-reactive protein [4], ferritin [5], creatinine [6], uric acid, hemoglobin, potassium, sodium, calcium, and glucose [7,8,9]. Moreover, given the growing evidence from human and animal studies of the involvement of both inflammation (local, innate immunity) and immune response (peripheral adaptive response) in ALS, markers of inflammation have also been evaluated. The neutrophil-to-lymphocyte ratio (NLR) has been recognized as a systemic marker of subclinical inflammation, and an increased ratio is of prognostic value in several disorders [10,11]. More recently, a few studies found that high NLR is associated with short survival duration in ALS [12,13]. We aimed to investigate whether NLR could be predictive of both disease progression and overall survival outcomes in ALS patients.

## 2. Materials and Methods

Patients were recruited during a multicenter study designed to explore, with a cross-sectional design, the association of some life-style factors with disease progression [14]. The study was conducted in two Italian centers (San Giovanni Rotondo and Modena), one in the Republic of Moldova (Chisinau), and one in Romania (Cluj-Napoca). Patients recruited in another center (Novara, Italy) were excluded from this analysis. The study was approved by the Institutional Review Boards of the coordinating center (N96/CE/2016) and the other centers. Written informed consent was obtained from all participants.

### 2.1. Patients

Patients of both sexes were consecutively enrolled from March 2016 to January 2020, in different periods in each center. The inclusion criteria were: (1) age higher than 18 years; (2) diagnosis according to the El Escorial criteria [15]; and (3) consecutive in- and out-patients with a new (incident) or already made (prevalent) clinical diagnosis of ALS. The exclusion criteria were: (1) patients with tracheostomy or receiving mechanical ventilation; (2) patients with percutaneous endoscopic gastrostomy; and (3) patients who did not sign the informed consent. Blood samples from the cubital vein were collected from subjects in the morning by venipuncture following overnight fasting (>8 h).

### 2.2. Data Collection and Outcomes Assessment

For each patient, we collected demographic (date of birth, gender, education) and clinical variables (date of onset and diagnosis, site of onset, diagnostic category according to El Escorial criteria, BMI, FVC, and therapy). Disease severity was estimated through the Amyotrophic Lateral Sclerosis Functional Rating Scale-Revised (ALSFRS-R) that evaluates the severity of the disease through a 12-item questionnaire [16]. The rate of disease progression (measured by the ΔFS score) at recruitment was calculated as the monthly decline of the ALSFRS-R score assuming a total score of 48 at onset: ΔFS = (48 − total ALSFRS-R at recruitment)/symptom duration in months [17]. The date of onset of disease was determined based on subjective complaints and information confirmed by relatives, and clinical charts. The NLR value was calculated as absolute peripheral neutrophil count divided by absolute periphery lymphocyte count. As for overall survival outcome, the patients’ follow-up was defined as the time between the date of recruitment and the occurrence of the tracheotomy or death, or the end of the follow-up (30 May 2021), whichever occurred first. At follow-up, the vital status of study patients was ascertained by the authors either by visit, telephonic interview with the patient or his/her relatives, or by linkage with the mortality registry office.

### 2.3. Validation Cohort

To validate our findings, we used data retrieved from the Pooled Resource Open-Access ALS Clinical Trials (PRO-ACT) Database (https://ncri1.partners.org/ProACT 30 october 2021) which includes information from 10,723 ALS patients who participated in industry, foundation, and academia sponsored clinical trials. The data available in the PRO-ACT database have been volunteered by the PRO-ACT Consortium members and is anonymized to protect patient privacy. Out of the patients included in the database (Appendix A), 708 were not eligible either because of missing (or incorrectly reported) follow-up time, vital status information, or both, 6397 because of missing information about neutrophil and lymphocyte count within 90 days prior or after the screening date, and 176 because of neurological comorbidity (i.e., Alzheimer’s disease, Parkinson’s disease, multiple sclerosis, and brain tumor). The final validation cohort consisted of 3442 eligible ALS patients.

### 2.4. Statistical Methods

The ALS patients’ characteristics were reported as mean ± standard deviation (SD) or as absolute and relative frequencies (%) for continuous and categorical variables, respectively. For continuous variables, the Q-Q plot test and the Shapiro–Wilk test were performed to assess the assumption of normality and, in case of non-normal distribution, the median along with interquartile range (IQR) was used instead of mean ± SD. Furthermore, in the presence of right-skewed distribution of continuous variables, statistical analyses were performed on log-transformed data. Comparisons of patients’ characteristics were performed with respect to NLR tertiles of the sample distribution. The association between a categorical clinical variable and NLR tertiles was assessed by the chi-square or the Fisher exact tests as appropriate, whereas the association between a continuous clinical variable and NLR tertiles was assessed by the ANOVA model. The patients’ mortality rate was computed as the number of deaths (occurred during the follow-up) over the total number of person-years. Comparison of mortality rates among NLR tertiles was assessed by a Poisson regression model. Kaplan–Meier survival curves were also shown, along with *p*-values from log-rank test. To assess the relationship between NLR values and overall survival, a random survival forest (RSF) [18] with 10,000 random decision trees (that split on NLR variable only) was performed, and variable dependence plot was produced, along with a loess smooth curve with shaded 95% confidence band. Furthermore, to detect the optimal cut-offs for NLR values, which identify patient subgroups at different mortality rates, a conditional inference tree (cTree) [19] was performed. Univariable and multivariable proportional hazard (PH) Cox regression models were performed to assess the unadjusted and covariate-adjusted relationship between the continuous or categorical NLR values and mortality risks, respectively. Results were reported as hazard ratios (HR) along with their 95% confidence interval (95% CI). To check the adequacy of fitted Cox regression models (i.e., checking of the functional form of a continuous covariate included in the Cox model and the assessment of the PH assumption), the Kolmogorov-type supremum test for functional form and for PH assumption were performed on the basis of 5000 data replicates (simulations) [19]. Adjusted survival curves were estimated from the multivariable Cox model, following the marginal weighting method [20]. Univariable and multivariable linear regression models were also performed to assess the unadjusted and covariate-adjusted relationship between the continuous (log-values) or categorical NLR values and progression rate (∆FS) (log-values), respectively. To assess the relationship between NLR and ΔFS values, a scatterplot with fitted regression line and shaded 95% confidence band was shown, along with the Pearson correlation coefficient (computed on log-values). Multivariable models were developed using the stepwise variable selection method (significance level for entry and staying in the model were 0.20 and 0.05, respectively) where NLR was forced to enter as the first covariate whereas the others were selected (by the stepwise method) among the following candidates: age at recruitment, gender, country, FVC, BMI, site of onset, and use of riluzole. A *p*-value < 0.05 was considered for statistical significance and was referred to two-tailed tests. All statistical analyses were performed using SAS Release 9.4 (SAS Institute, Cary, NC, USA). RSF, cTrees, adjusted survival curves, and all plots were performed using R Foundation for Statistical Computing (version 4.0, packages: survival, randomForestSRC, ggRandomForests, partykit, survminer, ggplot2).

## 3. Results

A total of 187 patients were initially recruited. We further excluded those who did not undertake a blood sample at recruitment (N = 18) and those for whom neutrophils and lymphocytes counts were unavailable within 3 months before or after recruitment (N = 23). The final eligible sample consisted of 146 patients (Figure 1), 88 men and 58 women, with a sex ratio of 1.5:1. No patient had clinical evidence of cancer, acute infection, or chronic active inflammatory disease. The median time elapsed between onset of disease and recruitment (i.e., disease duration) was 15 months (ranging from 2 to 30 months) and the median follow-up time was 2 years (IQR: 1–3). The mean ALSFRS-R score at recruitment was 35.8 ± 8.0 (ranging from 10 to 48), whereas the median progression rate was 0.66 (IQR: 0.26–1.10; ranging from 0 to 5.33).

Table 1 shows the patients’ characteristics overall and according to NLR tertiles. There were no significant differences as to gender, site of onset, El Escorial category, disease duration, FVC, BMI, and use of riluzole. Age at onset, at diagnosis, and at recruitment were significantly lower in the lowest NLR tertile. The ALSFRS-R score progressively decreased from the lowest to the highest NLR tertile. NLR values positively correlated with ALS progression rate (ΔFS) values (R = 0.28) (Figure 2A), and the median ΔFS was about 2.5 higher in the highest NLR tertile, compared to the lowest (Table 1). Furthermore, the mortality rate was 3-fold higher in the highest tertile (Table 1) and, as shown by the variable dependence plot produced from RSF, patients with higher NLR values had a worse survival than those with lower NLR values (Figure 2B). Survival rate was significantly lower for those in the highest tertile compared to the other two (Figure 2D). CTree identified the optimal cut-off in relation to the mortality rate at the NLR value of 2.315 (Figure 2C), which splits patients into two very different subgroups: the probability of 5-year survival for patients with an NLR > 2.315 was one third of that of patients with NLR ≤ 2.135 (Figure 2E).

To estimate the relationship between NLR values and the ALS outcomes, which were possibly affected by the presence of some clinical confounders, we performed univariable and multivariable regression and Cox models for ΔFS and overall survival outcomes, respectively (Table 2). The age at recruitment resulted in being the only relevant covariate, included in both models (i.e., significantly associated to both outcomes), which affected the NLR parameters’ estimates. As for ΔFS, the regression slope of NLR (log-values) passed from 0.60 (*p* < 0.001) before, to 0.49 (*p* = 0.006) after adjustment for age at recruitment. As for survival analysis (i.e., mortality), the HR estimated for each unitary increase of NLR (log-values) passed from 1.32 (95%CI: 1.16–1.50, *p* < 0.001) before, to 1.24 (95%CI: 1.08–1.41, *p* = 0.002) after adjustment for age at recruitment. Both the hypotheses of the linearity assumption about the NLR covariate form and the proportionality of the hazards cannot be rejected (*p*-values not significant). After adjustment for age at recruitment, higher NLR values still remained associated with a higher mortality risk when including NLR as a categorical variable, defined both by tertiles (data not shown) and by the cTree cut-off (Appendix A).

The clinical characteristics of the patients included in the validation cohort (selected from the PRO-ACT database) are shown in Appendix A and are reported overall and categorized by the same NLR tertile groups used in our sample. The clinical features of patients included in the validation cohort were similar to those of our sample, except for a slightly younger age, a higher disease duration, and a more frequent use of riluzole. We were not able to calculate the ΔFS for 83% of the subjects included in the validation cohort (ALSFRS-R information missing in 65% and disease duration missing in 20%). For this reason, we did not investigate the relationship between ΔFS and NLR in the validation cohort, but we limited the analysis of the association between NLR and mortality only. The HR estimated for each unitary increase of NLR (log-values) was 1.34 (95%CI: 1.16–1.54, *p* < 0.001), and it was 1.15 (95%CI: 0.98–1.36, *p* = 0.089) after adjustment for age at recruitment (Appendix A). The analysis of survival in the PRO-ACT database was in line with our findings, although the effect size was lower. Patients with an NLR > 2.315 (our NLR cut-off value) had a lower survival rate than those with NLR ≤ 2.135 (Appendix A, panel B), and survival was statistically different among NLR tertiles (Appendix A, panel A).

## 4. Discussion

In this study, we demonstrate that NLR evaluated in an early phase of the disease independently predicts survival in ALS and is associated with its progression rate. The two measures correlate, since the rate of symptoms progression is a predictor of survival [21]. NLR is emerging as an independent prognostic factor for several conditions, such as diabetes, chronic kidney disease, heart failure, hypertension or coronary artery disease, and some malignancies [11], but it has been evaluated in ALS only in two studies thus far [12,13]. Both studies divided NLR values in tertiles, as we did, finding a distribution of clinical predictors similar to our sample, including a higher age of onset and a lower ALSFRS-R score in the highest tertile, whereas sex, bulbar onset, diagnostic delay, FVC, and BMI were not different or did not show a trend. Progression rate was higher in the highest NLR tertile in both studies, and showed a trend towards lower values in the middle and lowest tertile. The Kaplan–Meier curves showed a pattern similar to our study, with overlapping survival in the lowest and middle tertiles, and a shorter survival in the highest one. The Cox multivariate regression analysis demonstrated that in our sample NLR was an independent predictor of ALS survival, even after adjustment for treatment and other possible prognostic factors. This was validated in an independent sample including a large number of ALS patients. Only age had a residual independent effect on survival. Age is positively associated with NLR in both sexes [11,22]: the higher the age, the higher the NLR; this is in agreement with the brain progressively acquiring an increased proinflammatory environment across the lifespan, a process termed ‘inflammaging’ [23].

NLR is simply a ratio whose increase reflects either an increase of numerator (neutrophils count) or a decrease of the denominator (lymphocytes count). The higher neutrophil count in ALS was associated with disease progression rate [24]. There is good evidence that peripheral myeloid cell populations are functionally altered in patients with ALS [25,26,27,28]. Neutrophils accumulate in the spinal cord both in animal models of ALS [29] and in ALS patients [30]. Accumulation of neutrophils in the spinal cord could induce damage through several mechanisms [31]. One possibility is that chronic neutrophil activation leads to reactive oxygen species production that exacerbates motor neuron degeneration [28] or alternatively they may contribute to modify microglial cells state. On the other side, the role of lymphocytes should be considered as well. In one study [32], the overall lymphocytes count was associated with better survival; another study found that the percentage of regulatory T (Treg) cells in the blood inversely correlated with ALS progression rate [33]. Among T lymphocytes, Tregs downregulate pro-inflammatory cytokine production, and directly inhibit interleukin (IL) 2 mRNA transcription. Tregs secrete anti-inflammatory cytokines (IL4, IL10, IL13) and neurotrophic factors, transform a Th1 to a Th2 response, and attenuate toxic microglial responses directly differentiating macrophages from M1 to M2 state [34]. Enhanced infiltration of Tregs and Th2L into the CNS of slowly progressing patients suggest that Tregs and Th2L could suppress inflammation in the spinal cord and influence the rate of ALS progression [35]. In this scenario, a low number of Treg cells may also contribute to the prognostic role of NLR.

Emerging evidence indicates that systemic inflammation may contribute to the progression of several neurodegenerative diseases. Inflammatory mechanisms including cell adhesion, neutrophil migration, lipid metabolism, and angiogenesis together with blood-brain barrier (BBB) breakdown have been associated with a more rapid cognitive decline in dementia [36]. Several findings suggest a BBB or blood-spinal cord barrier (BSCB) breakdown in ALS, favoring a role for inflammation. Increased CSF levels of albumin, IgG, and other blood-derived proteins have been reported in patients with ALS [37]. At a pathological level, brain and spinal cord tissue from patients with sporadic or familial forms of ALS exhibit fibrinogen, thrombin, IgG and hemosiderin deposits, pericyte degeneration [38], endothelial degeneration with reduced tight junctions, capillary basement membrane changes, and enlarged perivascular spaces [38,39], which are all signs of BBB/BSCB alteration. In a recent study on an animal model of ALS, elevated neuronal TAR DNA-binding protein 43 (TDP-43) levels induced microglial and astrocytic activation in the cortex, and exerted abundant immunoglobulin G, CD3, and CD4+ T cell infiltration as well as endothelial and pericyte activation, suggesting increased permeability of the BBB, thus enhancing the vulnerability of the cortex to the systemic inflammatory response [40], which in turn can initiate multiple pathways of neurodegeneration. In this light, the correlation between the use of riluzole and NLR that we observed in in the PRO-ACT cohort should be explored further.

Finally, blood NLR has been studied in association with gut microbiota, starting from the basis that both are associated with systemic inflammation. At the intestinal epithelial barrier level there is a strong crosstalk between gut microbiome, epithelial cells, and cells of the immune system, shaping the largest immunological organ in the body that has a central role in regulating immune homeostasis. Patients with a lower NLR had a greater diversity of gut microbiota [41], which has been taken into consideration as a possible factor modifying ALS progression both in animal models [42,43] and in humans [44]. All these results taken together may give a rational explanation why higher NLR values are associated with worse outcomes, although the underlying mechanism remains to be further examined.

Some limitations must be considered in our study: first, it is not population-based, thus it may be subject to patient selection; second, our sample was made by incident and prevalent cases, and the timing of NLR evaluation may play a role in driving its prognostic role. However, the median disease duration was 15 months, indicating that participants were still early in their ALS disease course at study entry. Third, we measured the NLR at a single time point during the course of the disease. It is unknown whether elevation of peripheral markers of inflammation, such as NLR, is an early sign or a later event. Data from AD and the general population tend to support the idea that systematic inflammation might be a later event during disease progression and aging [45]. Additional longitudinal studies with collection of samples at different time points are needed to understand whether NLR changes during the course of the disease, before using it to monitor peripheral inflammation in ALS. Fourth, our sample size and the post hoc nature of this study did not allow for sub-analyses by phenotype, genetic background, comorbidity, or other subgroups. Last, we can only speculate as to whether NLR is a mirror of some pathological process exacerbating progression, or it represents a secondary consequence of neuronal death (reverse causation).

In conclusion, the results of our preliminary and exploratory study confirm that NLR could be an independent marker of low-grade inflammation and immune response in ALS. Combining two metrics (high neutrophils and low lymphocytes count), NLR is a simple, rapid, inexpensive, easy-to-measure prognostic marker of disease progression and survival, and could be a candidate for use in clinical practice. However, these preliminary results must be tested in prospective studies, possibly population-based, with repeated measures of NLR, and in the context of the current clinical practice.

## Figures and Tables

**Figure 1 biomedicines-10-00354-f001:**
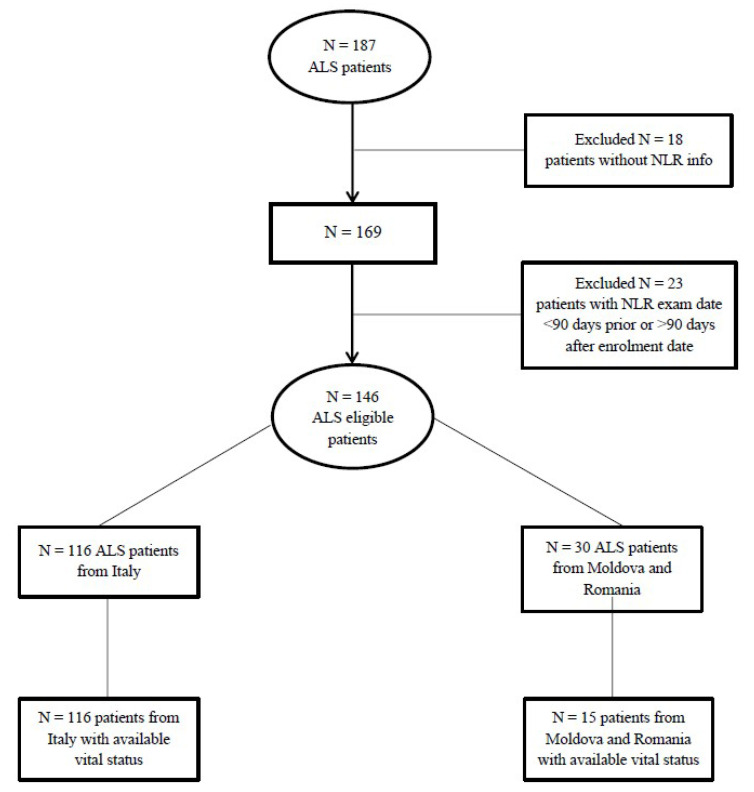
Patients’ disposition flow diagram.

**Figure 2 biomedicines-10-00354-f002:**
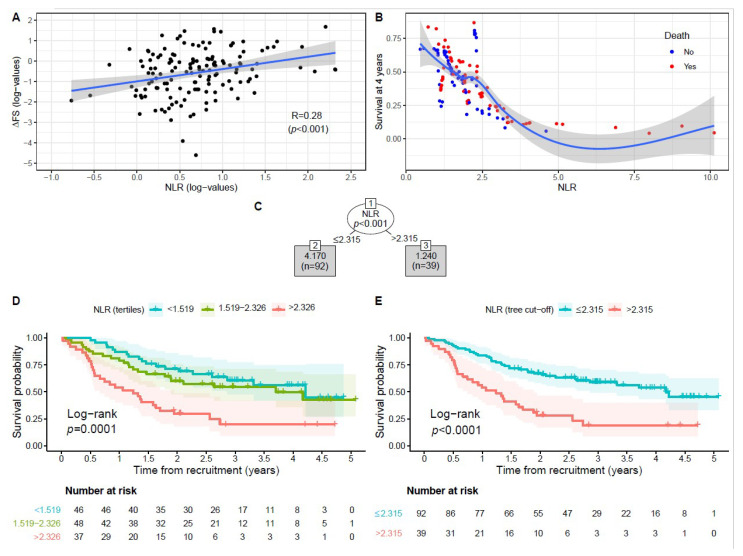
Relationship between neutrophil-to-lymphocyte ratio (NLR) values evaluated at recruitment and progression rate (ΔFS) and overall survival of ALS patients. (**A**) Log-transformed NLR and ΔFS values were shown by a scatterplot with fitted regression line, along with estimated Pearson correlation coefficient (R) and *p*-value; (**B**) variable dependence plot of patients’ survival at 4 years on NLR values estimated by the random survival forest algorithm with 10,000 trees. Individual cases are marked with blue (alive or censored) and red circles (dead). Loess smooth curve with shaded 95% confidence band indicates decreasing survival with increasing NLR values; (**C**) conditional inference tree (CTree) on NLR to predict the overall survival of ALS patients; (**D**,**E**) Kaplan–Meier (KM) survival curves according to NLR tertiles (**D**) or CTree groups (**E**). Censored observations are evidenced on the KM curves as tick marks (“+”). CTree identifies patient subgroups at different NLR mortality rate. The tree-growing algorithm recursively splits the data into subgroups, choosing the best binary split for NLR, to identify the most homogeneous sets within each node and the most heterogeneous ones between the nodes (i.e., NLR at 2.315 represents the optimal cut-off). Condition sending patients to left or right sibling is on relative branch. Grey squares (i.e., nodes 2 and 3) represent the final CTree classes. Numbers inside CTree classes represent the median survival time (in years, top) and the number of subjects (bottom), respectively. *p*-value from test of the global null hypothesis of independence between NLR groups and the response (i.e., patients’ overall survival) is reported in the root note (*p* = 0.001).

**Table 1 biomedicines-10-00354-t001:** Clinical characteristics of patients, overall and according to the tertiles of neutrophil-to-lymphocyte ratio (NLR) distribution evaluated at recruitment.

Variable	Category	All (N = 146)	I: NLR < 1.51 9(N = 48)	II: NLR [1.519–2.326] (N = 49)	III: NLR > 2.326 (N = 49)	*p*-Value	SMD	Missing Data (%)
Country—N(%)	Italy	116 (79.5)	36 (75.0)	45 (91.8)	35 (71.4)	0.029	0.364	0%
	Moldova/Romania	30 (20.5)	12 (25.0)	4 (8.2)	14 (28.6)			
Gender—N(%)	Females	58 (39.7)	22 (45.8)	20 (40.8)	16 (32.7)	0.407	0.181	0%
	Males	88 (60.3)	26 (54.2)	29 (59.2)	33 (67.3)			
Age at recruitment (years)	Mean ± SD	61.84 ± 10.85	58.38 ± 9.97	62.55 ± 11.01	64.53 ± 10.82	0.016	0.390	0%
Age at diagnosis (years)	Mean ± SD	60.99 ± 11.37	57.72 ± 10.06	61.21 ± 12.17	63.97 ± 11.12	0.024	0.380	0%
Age at onset (years)	Mean ± SD	59.78 ± 11.64	56.20 ± 10.41	59.99 ± 12.35	63.09 ± 11.25	0.013	0.410	0%
Education (years)	Mean ± SD	10.38 ± 4.36	11.12 ± 4.18	9.90 ± 4.73	10.12 ± 4.12	0.339	0.189	0%
Disease duration (months) ^§^	Median (IQR)	15.00 (9.00–29.75)	16.50 (10.00–36.25)	21.00 (9.00–39.00)	12.00 (9.00–23.00)	0.256 *	0.223 *	0%
Disease duration ^§^—N(%)	≤12 months	61 (41.8)	19 (39.6)	17 (34.7)	25 (51.0)	0.562 ^#^	0.362	0%
	13–24 months	40 (27.4)	11 (22.9)	14 (28.6)	15 (30.6)			
	25–36 months	15 (10.3)	6 (12.5)	5 (10.2)	4 (8.2)			
	37–48 months	7 (4.8)	3 (6.2)	3 (6.1)	1 (2.0)			
	>48 months	23 (15.8)	9 (18.8)	10 (20.4)	4 (8.2)			
Site of onset—N(%)	Bulbar	33 (22.6)	16 (33.3)	8 (16.3)	9 (18.4)	0.092	0.267	0%
	Spinal	113 (77.4)	32 (66.7)	41 (83.7)	40 (81.6)			
Escorial ALS—N(%)	Definite	51 (34.9)	12 (25.0)	15 (30.6)	24 (49.0)	0.076 ^#^	0.469	0%
	Possible	41 (28.1)	17 (35.4)	14 (28.6)	10 (20.4)			
	Probable	40 (27.4)	12 (25.0)	14 (28.6)	14 (28.6)			
	Suspected	14 (9.6)	7 (14.6)	6 (12.2)	1 (2.0)			
FVC—N(%)	<80%	50 (43.1)	12 (33.3)	19 (42.2)	19 (54.3)	0.202	0.286	20.5%
	≥80%	66 (56.9)	24 (66.7)	26 (57.8)	16 (45.7)			
	Missing values	30	12	4	14			
BMI (Kg/m^2^)—N(%)	<18.5	8 (5.5)	3 (6.2)	2 (4.1)	3 (6.1)	0.263 ^#^	0.318	0%
	18.5–24.9	70 (47.9)	18 (37.5)	23 (46.9)	29 (59.2)			
	≥25	68 (46.6)	27 (56.2)	24 (49.0)	17 (34.7)			
Use of riluzole—N(%)	No	90 (61.6)	32 (66.7)	28 (57.1)	30 (61.2)	0.626	0.131	0%
	Yes	56 (38.4)	16 (33.3)	21 (42.9)	19 (38.8)			
ALSFRS-R	Mean ± SD	35.77 ± 8.00	39.56 ± 4.99	35.20 ± 8.07	32.63 ± 8.89	<0.001	0.638	0%
ALS progression rate (ΔFS)	Median (IQR)	0.66 (0.26–1.10)	0.35 (0.18–0.93)	0.62 (0.25–1.09)	0.86 (0.53–1.92)	0.001 *	0.533 *	0%
Time from recruitment to last follow-up (years) ^§^	Median (IQR)	1.98 (1.03–2.98)	2.65 (1.57–3.41)	2.05 (1.16–3.04)	1.24 (0.53–2.04)	<0.001 *	0.648 *	10.3%
Mortality rate ^§^	events/PYs (rate per 100 PYs)	70/282 (24.8)	19/119 (15.9)	23/110 (20.9)	28/53 (52.8)	<0.001 °	---	10.3%

*p*-values from ANOVA models or chi-square statistics for continuous and categorical variables, respectively. ^#^ *p*-values from Fisher exact test; ° *p*-value from Poisson regression; * analysis on log-transformed values. NLR: neutrophil-to-lymphocyte ratio; ^§^ info available in 131 of 146 subjects only (see flow chart in Figure 1); SD: standard deviation; IQR: interquartile range (i.e., first-third quartiles); SMD: standardized mean difference (i.e., the average of all possible standardized mean differences); PYs: person-years.

**Table 2 biomedicines-10-00354-t002:** Association between neutrophil-to-lymphocyte ratio (NLR) values and both ALS progression rate (ΔFS) and mortality rate.

Outcome (Fitted Model)	Model Type	Variables Included into the Model (Covariates)	Covariates Type	Regression Coefficient (Slope)	*p*-Value	Groups (NLR Cutoffs) Comparison	HR (95%CI)	*p*-Value	Test For Functional Form (*p*-Value) ^#^	Test for PH (*p*-Value) ^#^
ΔFS (log-values *) (linear regression)	Univariable (unadjusted)	NLR (log-values *)	Continuous	0.602	<0.001	--	--	--	--	--
	Multivariable ^°^ (adjusted)	NLR (log-values *)	Continuous	0.490	0.006	--	--	--	--	--
		Age at recruitment (years)	Continuous	0.020	0.019					
Mortality rate (Cox model)	Univariable (unadjusted)	NLR	Continuous	--	--	--	1.32 (1.16–1.50)	<0.001	0.264	0.866
		NLR (tertiles)	Categorical	--	--	(1.519–2.326) vs. <1.519	1.31 (0.71–2.41)	0.384	--	0.874
				--	--	>2.326 vs. <1.519	3.13 (1.74–5.63)	<0.001	--	0.433
		NLR (tree-based cut-off)	Categorical	--	--	≤2.315 vs. >2.315	2.67 (1.65–4.31)	<0.001	--	0.742
	Multivariable ^°^ (adjusted)	NLR	Continuous	--	--	--	1.24 (1.08–1.41)	0.002	0.430	0.818
		Age at recruitment (years)	Continuous	--	--	--	1.06 (1.04–1.09)	<0.001	0.443	0.724
		NLR (tertiles)	Categorical	--	--	(1.519–2.326) vs. < 1.519	1.03 (0.55–1.91)	0.934	--	0.787
				--	--	>2.326 vs. <1.519	2.37 (1.29–4.35)	0.005	--	0.712
		Age at recruitment (years)	Continuous	--	--	--	1.06 (1.04–1.09)	<0.001	0.257	0.712
		NLR (tree-based cut-off)	Categorical	--	--	≤2.315 vs. >2.315	2.16 (1.32–3.53)	0.002	--	0.939
		Age at recruitment (years)	Continuous	--	--	--	1.06 (1.04–1.09)	<0.001	0.471	0.680

HR: hazard ratio; CI: confidence interval; PH: proportional hazards; NLR = Neutrophil-to-Lymphocyte Ratio. * Continuous NLR and progression rate (ΔFS) values were log-transformed because of the right-skewed distribution of their original values. ^#^ To check the adequacy of fitted Cox regression models (i.e., checking of the functional form of a continuous covariate included into the Cox model and the assessment of the PH assumption), the Kolmogorov-type supremum test for functional form and for PH assumption was performed on the basis of 5000 data replicates (simulations). As no polynomial (and interaction) terms of each continuous covariate were included into each Cox model, the test for functional form assessed whether a linear relationship existed between a one unit increase of the continuous covariate and the risk of death. ^°^ Multivariable models were developed using the stepwise variable selection method (significance level for entry and staying in the model were 0.20 and 0.05, respectively) where NLR was forced to participate as the first covariate whereas the other covariates were selected (by the stepwise method) among the following candidates: age at recruitment, gender, country (Italy vs. Moldova/Romania), FVC, BMI, site of onset, and use of riluzole.

## Data Availability

The data presented in this study are available on request from the corresponding author. The data are not publicly available due to ethical and privacy restrictions.

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
