# Peer review of "Neutrophils-to-Lymphocyte Ratio Is Associated with Progression and Overall Survival in Amyotrophic Lateral Sclerosis"

_biomedicines, 2022, doi:10.3390/biomedicines10020354_

Round 1

Reviewer 1 Report

Leone et al. paper is describing a very interesting observation about correlation of NLR with progression and overall survival in ALS patients. To find such an easy and cheep biomarker of ALS prognosis would as Holy Grail finding and KM curves look very promissing in the patient´s cohort, unfortunately, contrary to data from database. However, the study has fatale problems wich makes unable to publish that in current form.

1/ NLR is dynamic and could be changed fast. One static evaluation is not possible to cover all situations related to acute inflammation. Series of NLR must be provided and all potential source of inflammatory changes must be included not only stated (CRP, cultivations, procalcitonin level).

2/ NLR could be changed in different age groups and age range is very broad to test on such a small patient group the plausibility of hypothesis. Moreover, there is a significant age difference between tertiles in the patient´s cohorte contrary to database validation cohort. 

3/ NLR should be evaluated at the similar time in the course of the disease, which is in the study only poorly defined, because, as authors discussed, it could be changed during disease course. 

4/ NLR could be influenced by microbiome, as mentioned in the discussion. There are missing data about microbiome, moreover, it could be expected, that microbiome of patients from Italy is strongly different than Moldovian and Romanian.

5/ NLR and microbiome are associated with C9orf72 status. There is no data about genetics

6/ There are not data about chronic comorbidities potentially influencing NLR (cancer, AI disorders) neither in patient´s cohort nor database group.

7/ In patient´s cohort, there is any information about dementia/cognitive deterioration, which is exclusion criterion in the database group. Moreover, very strange is only very limited dementia/comorbit AD cognitive profile in database patients since cognitive problems of different degree are seen up to 40% of ALS patients. Are there in database patients verified by autopsy? And, thus, how many patients from patient´s cohort were autopsied?

8/ 146 patients, moreover, very heterogenous, are very limited in number for multiparameters evaluation. Some very interesting observations are in contrary to robust data from database and could be, probably, not significant on the larger series.

9/ Very interesting point is Riluzole administration and NLR in validation cohort. It is not discussed at all.

Finally, paper is very interesting, well written, taking very urgent need to verification of some good biomarker for ALS prognosis of course progression and survival. In the current form is, however, not possible to publish, since the study is only preliminary, with small number of heterogenous patient, with prominent methological problems, namely not beeing population-based.

Reviewer 2 Report

Leone et al. present an interesting study on the potential relationship between the neutrophils-to-lymphocytes ratio (NLR) and the prognosis of ALS patients. It is noteworthy the high number of patients screened. Nevertheless, this referee has several concerns and questions regarding this study:

  1. In the Abstract the authors state “Possible prognostic serum biomarkers include albumin, C-reactive protein (CRP), ferritin, creatinine, uric acid, and the neutrophil-to-lymphocyte ratio (NLR), a marker of subclinical inflammation”. However, other potential biomarkers are missing i.e. hemoglobin, glucose, K+, Na+ and Ca2+. E.g., in a recent study run by the Karolinska Institute, it was shown that serum creatinine, albumin, CRP and glucose measured at the time of ALS diagnosis as well as their temporal changes could serve as prognostic biomarkers for ALS (DOI: 10.1111/ene.14409).
  2. The study concludes that the mortality rate passed from 15.9 events per 100 person-years in patients belonging to the lowest NLR tertile, to 52.8 in those in the highest tertile. This seems quite clear in supporting the NLR as a prognostic biomarker. However, this referee has a number of concerns regarding how patients have been recruited (Results section): The median time elapsed between onset of disease and recruitment (i.e. disease duration) was 15 months (ranging from 2 to 30 months) and the median follow-up time was 2 years (IQR: 1-3). The mean ALSFRS-R score at recruitment was 35.8±8.0 (ranging from 10 to 48), whereas the median progression rate was 0.66 (IQR: 0.26-1.10; ranging from 0 to 5.33). Groups should avoid so different ranging, such as 2 to 30 months between onset and recruitment or an ALSFRS-R score from 10 to 48. The clinical status of patients is too different to be compared within the same group. Data would be more reliable by grouping the patients in a different way. And, consequently, the entire study recalculated accordingly.
  3. Taking into account the multiple factors that can affect, in a healthy person, the neutrophil/lymphocyte ratio; it should be convenient include more than one value for each patient. If more of one value is available in the medical report of the patients, criteria used for acceptance or rejection of the data, would increase the significance of the means, and give greater strength to the results obtained.
  4. To this referee it is unclear which was the rational to determine that the optimal cut-off value which best classified patients with the lowest and the highest mortality rate was set at the NLR value of 2.315. These criteria should be much better explained.

Round 2

Reviewer 1 Report

The authors improved the paper, however some point must be clarify more.

Altough present form is taken as preliminary data needed to be confirmed, this must be more visible for the readers. 

I don´t understand why this study is not designed as single-centre, heterogeneity of three centres from other geo-cultural background would provide more questions as I mantioned previosly (microbiota), moreover, the principles of diagnostics could be different as well as lab tests.

Different concomitant comorbid diseases should be examine more deeply. The presence of second or event third disease is very frequent and this question is not answered. Potential difference NLR in these situation would be of interest.

Author Response

We wish to thank the reviewer for his/her valuable comments, that helped to improve the paper.

  • Altough present form is taken as preliminary data needed to be confirmed, this must be more visible for the readers.

       Response:  According to reviewer’s comment, we have further stressed the very preliminary and exploratory nature of our study in the discussion section. 

  • I don´t understand why this study is not designed as single-centre, heterogeneity of three centres from other geo-cultural background would provide more questions as I mantioned previosly (microbiota), moreover, the principles of diagnostics could be different as well as lab tests.

Response: The study was designed as a multicenter study for evaluating the impact of smoking, and alcohol, tea and coffee consumption on ALS progression.  The evaluation of the association of NLR with outcomes is a post-hoc analysis of these already collected data.  Although we fully understand and agree with the reviewer that a larger study could give more solid information, still we decided to publish these preliminary data, given its novelty.  We are planning a larger study, taking into account all the points raised by the reviewer. 

  • Different concomitant comorbid diseases should be examine more deeply. The presence of second or event third disease is very frequent and this question is not answered. Potential difference NLR in these situation would be of interest.

ResponseWe agree with the reviewer on the potential interest of examining co-morbidity.  Being designed to test the hypothesis of the association between some lifestyle habits and ALS progression, a systematic collection of comorbidities was not carried on.  For this reason, a full analysis of comorbidities is not possible.  As already listed in the results section, no patients had acute infection or chronic active inflammatory disease. As requested by the referee in his/her first revision, we have added cancer to this list.  We have added among the limitations the lack of an analysis by subgroup with and without co-morbidity.  

Reviewer 2 Report

It is an interesting study on the potential relationship between the neutrophils-to-lymphocytes ratio (NLR) and the prognosis of ALS patients. After carrying out the review of the work, my impression is your response is sufficiently satisfactory.

Author Response

there are no further comments by the second referee